# Thermally Modified Wood Exposed to Different Weathering Conditions: A Review

**Delfina Godinho** [1,2,*], **Solange de Oliveira Araújo** [1], **Teresa Quilhó** [1], **Teresa Diamantino** [2] **and Jorge Gominho** [1]

1 Centro de Estudos Florestais, Instituto Superior de Agronomia, Universidade de Lisboa, Tapada da Ajuda, 1349-017 Lisboa, Portugal; araujo@isa.ulisboa.pt (S.d.O.A.); terisantos@isa.ulisboa.pt (T.Q.); jgominho@isa.ulisboa.pt (J.G.)
2 Laboratório Nacional de Energia e Geologia, I.P. (LNEG), Estrada do Paço do Lumiar, 22, 1649-038 Lisboa, Portugal; teresa.diamantino@lneg.pt
* Correspondence: delfina.godinho@lneg.pt

**Abstract:** Outdoor wood applications are exposed to several different biotic and abiotic factors, and for that reason, they require protection to increase their service life. Several technologies of wood protection are already commercialized. One of these technologies is thermal modification, which refers to the structural, mechanical, and chemical transformations occurring in the lignocellulosic material when gradually heated up to specific temperature ranges. In the past few years, several researchers have undertaken weathering resistance evaluations on different wood species. Some cases have considered natural exposure in different countries with different climatic conditions, while others focused on artificial exposure under UV and xenon radiation tests. Most works evaluated the weathering effects on the chemical, mechanical and physical, and anatomical shifts compared to the original characteristics of the material. This review has established a considerable lack of studies in the bibliography focusing on abiotic factors, such as the industrial and maritime environment, or even isolated climatic factors such as salt spray (simulating maritime environments) or pollutant gases (simulating industrial environments). This lack of information can be an opportunity for future work. It could help to understand if thermally modified wood is or is not sensitive to pollutant gases or salinity, or to a combination of both. By knowing the degradation mechanisms caused by these factors, it will be possible to study other forms of protection.

**Keywords:** degradation; abiotic and biotic factors; thermal modification; weathering; wood

## 1. Introduction

### 1.1. Context

Wooden constructions are usually characterized by different components and treatments that together offer the best possible properties of load-bearing capacity, thermal, acoustic, moisture insulation, fire resistance, and long service life. Increasing the proportion of wood in constructions can reduce the quantities needed of other construction materials, such as concrete, steel, and bricks. These construction materials do not come from renewable raw materials. They require a great deal of energy for their production and entail higher carbon dioxide emissions [1].

There are a wide variety of applications of wooden structures, and nobody questions the value of the majority of these products [2,3] for building materials, including indoor and outdoor products, structural frames, window and door frames, floors, and façade systems [4–6]. However, wood is a complex biological tissue composed of different cell types [7]. This cellular diversity, with a diverse molecular structure, largely determines the physical and mechanical properties and profoundly influences the performance of wood as a construction material and its demand as such [8]. Therefore, knowledge of the wood structure and species identification is essential [1,9].

The world's political and economic decisions are increasingly determined by resource and energy scarcity, with climate change a key topic concerning fossil energy consumption. The forest sector and wood-based industries are challenged by this trend and tend to participate favorably in the sustainability debate [4] and its contribution to the global economy. The wood sector has shown it can develop competitive and innovative industrial applications with an appropriate service life [10,11]. Recently, some authors have studied the potential of mid-rise urban buildings designed with engineered timber, to provide long-term carbon storage and avoid the carbon-intensive production of mineral-based construction materials [6].

### 1.2. Exterior Wood

For outdoor use, wood is exposed to many abiotic and biotic factors, such as weather conditions and biodegradation, which negatively affect the wood material's physical, biological, chemical, and mechanical properties [12].

These abiotic and biotic factors can compromise wood's mechanical and physical properties, particularly the combination of different weather conditions (humidity, temperature, solar irradiation, salinity variations) and biodegradation (xylophages and fungus) [13–16]. With climatic changes, extreme drought scenarios are predicted that can cause UV damage to the surface, due to excessive sunlight. Thus, evaluating its durability is critical to predicting the negative impacts of outdoor exposure on wood. It also helps predict the lifetime and the financial cost of scheduled maintenance.

### 1.3. Wood Modification

Due to the environmental problems associated with chemicals, many industries, including the wood industry, are looking for environmentally friendly wood treatments to improve wood durability.

Several wood treatments are used for wood protection, depending on its final use. Chemical modifications of the solid wood allow the improvement of dimensional stability, mechanical properties, or resistance to biodegradation [17]. These modifications can be achieved using chemicals or by thermal conversion and can help to strengthen the wood surface and its structural behavior [18,19].

Several technologies of wood modification are already on the market and demonstrate the potential of these modern technologies. One of them is thermal modification, as presented in the review performed by Sandberg et al. [18,20] and Jones et al. [21].

This present review aims to understand what kind of weathering studies on thermally modified wood have been undertaken in recent years.

## 2. Wood Modification

Thermally modified wood (TMW) is a material derived from a treatment that combines temperature and moisture, avoiding harmful substances while providing better energy efficiency and drying quality [2]. Such types of processes can considerably improve the performance of timber in several aspects. The treatment is usually achieved at temperatures between 120 °C and 260 °C, depending on the industrial process and desired end-product characteristics (Table 1).

This kind of treatment is widely used, established by several company trademarks and patents around the globe [19–21]. The first process we studied was carried out by Burmester, who studied the effects of temperature, pressure, and moisture in a closed system, then named Feuchte-Wärme-Druck (FWD) [18]. Throughout this technology development, several commercial processes were created, such as Lignostone® and Lignifol® in Germany, and Staypak® and Staywood® in the United States of America. More recently, other commercial methods were introduced in Europe: the Thermowood® process in Finland, the Plato® process in The Netherlands, and the Perdure® process and Retification® in France [22–25]. During the 2000s, other thermal modification processes were created: those using vegetable oils, such as OHT® (oil heat treatment) and those using a vacuum system,

such as VacWood® (Thermo vacuum-treated wood) [18]. In Europe, the ThermoWood process is the one most commonly used commercially [18]. Table 2 shows the wood species that are widely used in Finland and other countries.

**Table 1.** Different thermal modification industrial processes (adapted from Sandberg et al. [18]).

| Process | App. Year | Temperature (°C) | Process Duration (h) | Pressure (MPa) | Atmosphere | System Type |
|---|---|---|---|---|---|---|
| FWD | 1979 | 120–180 | ≈15 | 0.5–0.6 | Steam | Closed system |
| Plato | 1980 | 150–180/ 170–190 | 4–5/70–120 up to weeks | Super atmospheric pressure (partly) | Saturated steam/ heated air | A four-stage process |
| ThermoWood | 1990 | 130/185– 215/80–90 | 30–70 | Atmospheric | Steam | Continuous steam flow through the wood under processing removes volatile degradation products. |
| Le Bois Perdure | 1990 | 200–230 | 12–36 | Atmospheric | Steam | The process involves drying and heating the wood in steam. |
| Retification | 1997 | 160–240 | 8–24 | - | Nitrogen or other gas | The nitrogen atmosphere guarantees a maximum oxygen content of 2%. |
| OHT | 2000 | 180–220 | 24–36 | - | Vegetable oils | Closed system |

**Table 2.** Wood species studied for commercial purposes [25].

| Softwood Species | Hardwood Species |
|---|---|
| Pine (*Pinus sylvestris*) | Birch (*Betula pendula*) |
| Spruce (*Picea abies*) | Aspen (*Populus tremula*) |
| Radiata pine (*Pinus radiata*) | Ash (*Fraxinus excelsior*) |
| Maritime pine (*Pinus pinaster*) | Larch (*Larix sibirica*) |
| | Alder (*Alnus glutinosa*) |
| | Beech (*Fagus sylvatica*) |
| | Eucalyptus (*Eucalyptus* sp.) |

Thermal modification treatments alter the structure and chemical composition of the wood cell walls [26–28]. These changes are responsible for modifying physical and mechanical properties. The main effects are improvement of dimensional stability, reduction of hygroscopicity (due to a decrease in the equilibrium moisture content and wettability), and the improvement of resistance to biological attack [19,23]. On the downside, these treatments can cause a reduction in some mechanical and physical properties, namely, in the modulus of elasticity (MOE), the modulus of rupture (MOR), impact toughness, abrasion resistance, hardness, and roughness [23,29–32], depending on the wood species (Table 3).

According to the literature, different reactions occur during thermal treatments [36]. Figure 1 summarizes the chemical changes that occur during thermal modification. Even at lower temperatures, hemicelluloses are the first structural components affected by depolymerization and hydrolysis. Cellulose is the next to be affected: hydrolysis and recrystallization of the amorphous region increase the crystallinity index in cellulose [18,25,37]. Finally, lignin reductions occur, with homolytic cleavage and polycondensation as the main reactions. Other reactions, such as extractive flow from inside the wood and a decrease in pH, have also been reported [25].

**Table 3.** Examples of the mechanical properties of some wood species before and after the thermal modification.

| Species | Thermal Modification | Time | MOE (MPa) | MOR (MPa) | Roughness (μm) | Hardness (kg) | Reference |
|---|---|---|---|---|---|---|---|
| Ash (*Fraxinus excelsior*) | Control | 90 min in steam atmosphere | 7760 (850.94) | 90.68 (5.78) | - | - | [33] |
| | 212 °C | | 9990 (1838.94) | 74.04 (7.59) | | | |
| Iroko (*Milicia excelsa*) | Control | 90 min in steam atmosphere | 11960 (1719.88) | 121.90 (18.85) | - | - | |
| | 212 °C | | 12860 (960.73) | 114.83 (16.14) | | | |
| Scots pine (*Pinus sylvestris*) | Control | 90 min in steam atmosphere | 9644 (498.33) | 89.54 (7.45) | - | - | |
| | 190 °C | | 8808 (1219.58) | 74.18 (9.77) | | | |
| Spruce (*Picea orientalis*) | Control | 90 min in steam atmosphere | 7618 (320.66) | 75.20 (3.00) | - | - | |
| | 190 °C | | 8985 (1244.55) | 72.50 (8.92) | | | |
| Black Alder (*Alnus glutinosa*) | 190 °C | 3 h | - | - | 36.08 (1.5) 35.35 (1.6) | 341.6 (26.1) 361.29 (24.3) | [31] |
| Red Oak (*Quercus rubra*) | 190 °C | 3 h | - | - | 57.82 (6.5) 54.28 (3.1) | 662.00 (73.7) 533.72 (38.3) | |
| Southern Pine (*Pinus taeda*) | 190 °C | 3 h | - | - | 27.16 (1.4) 27.00 (1.4) | 263.44 (28.4) 270.97 (29.5) | |
| Yellow Poplar (*Liriodendron tulipifera*) | 190 °C | 3 h | - | - | 44.08 (1.8) 44.01 (1.4) | 352.34 (50.8) 354.65 (46.3) | |
| Maritime pine (*Pinus pinaster*) | Control | 2 h | 1110 (13.5%) | 130 (21.5%) | - | - | [34] |
| | 200 °C | | 1130 16.4%) | 127 (17.4%) | | | |
| | 240 °C | | 1070 (17.2%) | 104 (12.2%) | | | |
| | 260 °C | | 1130 (34.9%) | 76 (24.2%) | | | |
| | 300 °C | | 7800 (<2%) | 51 (11.0%) | | | |
| Eucalyptus (*Eucalyptus globulus*) | Control | 2 h | 1440 (5.5%) | 129 (5.9%) | - | - | |
| | 200 °C | | 1580 (14.9%) | 105 (21.7%) | | | |
| | 240 °C | | 1260 (14.9%) | 86 (25.6%) | | | |
| | 260 °C | | 1410 (8.6%) | 91 (13.9%) | | | |
| | 300 °C | | 4600 (< 2%) | 28 (8.2%) | | | |
| Beech (*Fagus sylvatica*) | Control | 2 h | 1190 (24.0%) | 146 (26.6%) | - | - | |
| | 200 °C | | 1230 (18.5%) | 167 (6.5%) | | | |
| | 240 °C | | 9600 (25.8%) | 124 (11.5%) | | | |
| | 260 °C | | 1040 (20.2%) | 105 (21.6%) | | | |
| | 300 °C | | 8140 (<2%) | 52 (12.0%) | | | |
| Acacia (*Acacia melanoxylon*) | Control | 2 h | 1610 (4.1%) | 138 (6.7%) | - | - | |
| | 200 °C | | 1640 (7.5%) | 141 (2.8%) | | | |
| | 240 °C | | 1040 (14.9%) | 83 (9.7%) | | | |
| | 260 °C | | 1300 (5.9%) | 82 (2.8%) | | | |
| | 300 °C | | 8400 (13.8%) | 47 (28.9%) | | | |
| Oak (*Quercus faginea*) | Control | 2 h | 1130 (9.9%) | 102 (9.2%) | - | - | [34] |
| | 200 °C | | 1150 (13.3%) | 91 (12.6%) | | | |
| | 240 °C | | 1090 (16.4%) | 83 (20.2%) | | | |
| | 260 °C | | 1120 (5.2%) | 74 (3.0%) | | | |
| | 300 °C | | 1010 (12.8%) | 68 (12.8%) | | | |
| Pedunculate Oak (*Quercus robur*) | Control | | 11731 (4219) | | - | - | [35] |
| | 160 °C | | 11021 (350) | | | | |
| | 180 °C | | 10846 (1168) | | | | |
| | 200 °C | | 11639 (1028) | | | | |

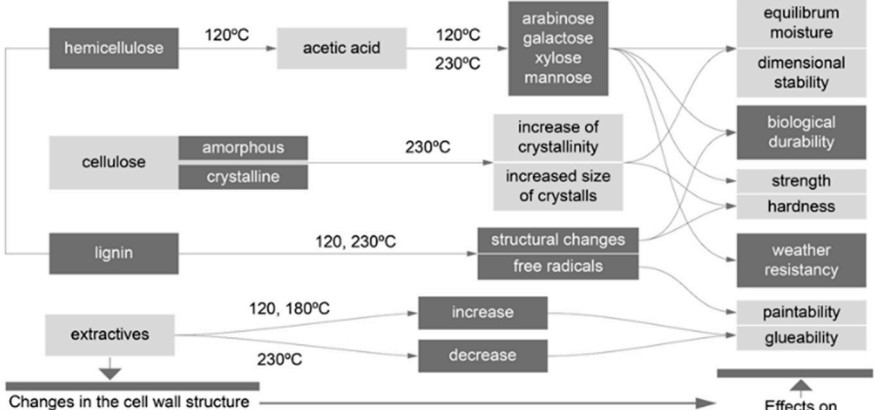

**Figure 1.** Chemical changes that occur during thermal modification (adapted from the International Table 2003 [25]).

The wood anatomy is also affected. Fengel and Wegener [38] analyzed thermo-modified spruce wood treated at 150 °C by electron microscopy (SEM) and noticed changes in the cellular structure (cracks between the S1 and S2 layers in the corner cells). Other studies reported the destruction of tracheid walls, ray tissues, pit deaspiration, small cracks between tracheids, radial cracks, and transverse ruptures [26,27,39]. Figure 2 shows SEM images depicting the changes in the cellular wood structure of yellow poplar (*Liriodendron tulipifera*) and Korean red pine (*Pinus densiflora*) under different heat-treatment temperature conditions [28].

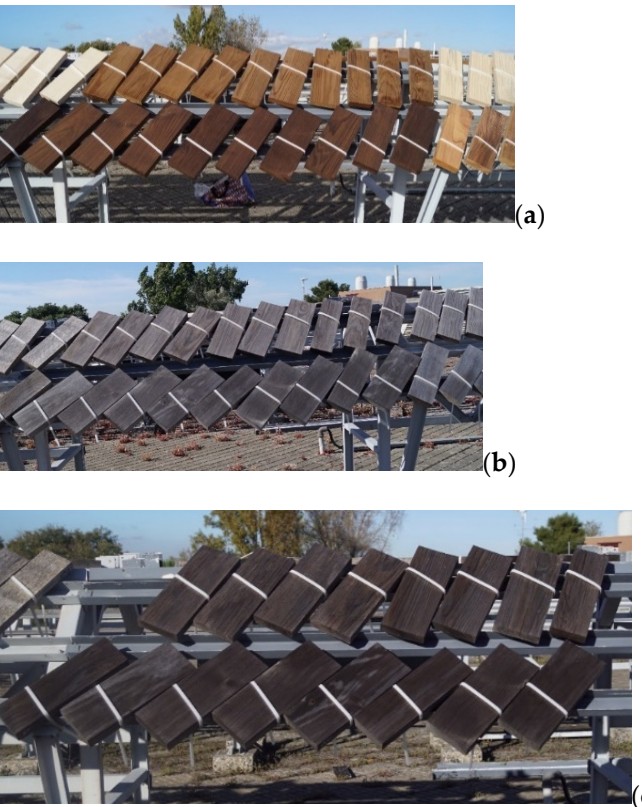

**Figure 2.** Pictures of color evolution during weathering exposure: (**a**) 0 months of exposure; (**b**) 9 months of exposure; (**c**) 12 months of exposure.

## 3. Weathering

Weathering is a term used to define the slow degradation of any material exposed to the weather. In the natural world, different weathering agent combinations can be found: sunlight, moisture gradients, temperature changes, chemical agents, atmospheric pollution, abrasion promoted by windblown particles, human activity, and biological agents [40–42].

There is a difference between wood weathering and wood decay. Weathering primarily affects the wood surface, as well as the chemical structure and extractives. The main factor for these changes caused by sunlight is the UV radiation part of the light spectrum. It promotes photodegradation, resulting in color changes that, over time, make the wood greyish or even silver, which can be esthetically valuable in some situations. Another consequence of UV radiation is the degradation of lignin and carbohydrates [40]. Weathering also causes surface erosion due to wind and particle abrasion [40]. Wood decay affects all thickness and bulk in the wood [37]. Fungi and xylophages (or both) cause wood decay since they contribute to biodegradation [40,42]. Wood decay only occurs when the wood has free water. Air humidity and rain can cause it; it also occurs when the wood is not dried correctly or is in direct contact with soil [43–46].

One of the biggest challenges when using wood is its performance during weathering because it can be strongly affected. This usually happens long before the material's mechanical properties are compromised, so the esthetics of wood is often the crucial determinant of its service life. The rapid change in appearance puts wood at a disadvantage in comparison with more weather-resistant materials. Lignin is the main component responsible for the low light-stability of wood. Its degradation also facilitates fungal colonization, leading to fungal staining, which causes the grey discoloration of wooden surfaces outdoors [45,46]. Understanding weathering mechanisms, the role of the altering factors, and their effect on the anatomical, chemical, and physical properties of wood is fundamental to assessing timber artifacts under actual conditions. It is also crucial to predict future performance and, possibly, to ensure long-term preservation and maintenance [47–49]. These requirements need some outdoor testing to understand the degradation mechanisms of different wood species under other climatic conditions [5,50].

In the last years, several studies have been conducted to evaluate the impact of weathering in thermally modified wood. Studies on the exposure of wood to natural weathering conditions (urban environments) [5,12,42,51–56] or artificial exposure through the use of climatic chambers (UV radiation, temperature, and humidity) [57–62] were conducted.

These natural or artificial conditions have also been used to evaluate the wood surface changes (color and roughness) and the alteration of mechanical properties (the modulus of elasticity and modulus of rupture) (Table 4) [33,52,54,57].

**Table 4.** Example of wood species surface and mechanical properties changes (adapted from Tomak et al. [33]).

| Species | Thermal Modification | Time of Exposure (Months) | Roughness (μm) | Color (CIE*Lab) | | | | MOR (MPa) | MOE (MPa) |
| --- | --- | --- | --- | --- | --- | --- | --- | --- | --- |
| | | | | ΔL* | Δa* | Δb* | ΔE | | |
| Ash (*Fraxinus excelsior*) | 190 °C for 90 min | 0 | 34.72 (6.81) | −37.57 (2.17) | 2.73 (0.27) | −5.61 (0.65) | 38.09 (2.17) | 74.04 (7.59) | 9990 (1838.94) |
| | | 24 | 41.40 (9.29) | 14.91 (3.81) | −6.60 (1.70) | −9.66 (3.11) | 19.55 (3.16) | 72.29 (9.14) | 11845 (1775.12) |
| Iroko (*Chlorophora excelsa*) | 190 °C for 90 min | 0 | 36.81 (7.11) | −6.81 (0.95) | 1.05 (0.42) | −3.07 (0.86) | 7.61 (0.82 | 114.83 (16.14) | 12860 (960.73) |
| | | 24 | 52.08 (6.05) | 11.35 (5.33) | −9.14 (0.55) | −16.61 (1.30) | 21.70 (3.10) | 91.09 (10.65) | 10049 (1666) |
| Scots pine (*Pinus sylvestris*) | 212 °C for 90 min | 0 | 32.57 (4.12) | −25.69 (2.45) | 6.04 (0.63) | 2.35 (0.75) | 26.51 (2.31) | 74.18 (9.77) | 8808 (1219.5) |
| | | 24 | 48.75 (8.84) | 3.54 (1.11) | −10.42 (1.15) | −20.98 (2.42) | 23.61 (2.46) | 66.01 (9.64) | 7487.14 (743.34) |
| Spruce (*Picea orientalis*) | 212 °C for 90 min | 0 | 33.64 (7.30) | −27.94 (2.78) | 7.27 (0.72) | 4.52 (1.07) | 29.20 (2.83) | 72.50 (8.92) | 8985 (1244.55) |
| | | 24 | 45.71 (8.47) | 2.22 (2.71) | −10.39 (1.08) | −21.56 (2.26) | 24.13 (2.27) | 70.60 (9.94) | 7726 (1066.98) |

Figure 2 shows the color evolution during time.

### 3.1. Natural Weathering

Natural weathering modifies the molecular structure of wood through a complex combination of chemical, mechanical, biological, and light-induced changes that coincide and affect each other [41].

Several studies evaluated different thermally modified wood species (with treatments different from those indicated above) experiencing weathering under urban environments. The main environmental parameters that were monitored in test sites for all the studies were solar radiation, temperature, and relative humidity.

Nuopponen et al. [52] used *Pinus sylvestris* in their study, and the wood was treated at VTT Building and Transport, Finland, using the Thermowood® process. The panels (both thermally modified and unmodified) were weathered vertically for seven years (1994–2001) in Espoo (Finland). The modified wood's thermally weathered surface was still rich in aromatic and conjugated carbonyl structures when inspected via spectroscopic analyses. On the other hand, unmodified weathered samples were enriched with cellulose. In addition, the lignin in weathered unmodified samples leached more quickly than in treated wood samples. Modified wood generally showed better weathering behavior because its structure and degradation products did not leach as easily as in unmodified samples.

Another study from Lesar et al. [56] analyzed the performance of façade elements made of five different thermally modified wood species in a model house. The built-in form replicates all positions that wood can be applied in construction (façades and decks). The wood species studied were Norway spruce (*Picea abies*), European larch (*Larix decidua*), common beech (*Fagus sylvatica*), linden (*Tilia* sp.), and ash (*Fraxinus excelsior*). All wood species underwent thermal modification using a commercial procedure (Silvapro®).

The authors evaluated the changes in color and moisture content during a two-year exposure period in Ljubljana (Slovenia). They concluded that blue-stain fungi changes in color and growth on façade and decking correlated with solar radiation and water condensation. Unmodified wood underwent a greater color change than modified wood, but modified wood became grey faster than unmodified wood. They also noticed that decking of modified wood had a higher moisture content than that of unmodified wood, except for modified larch, which was altered at lower temperatures. The authors theorized that this happened because thermally modified wood has a higher permeability [56]. Finally, after the treatment, the formation of microcracks occurs, as well as degradation from tylosis.

Ugovšek et al. [5] evaluated the weathering resistance of wooden windows and façade elements made from thermally modified and unmodified Norway spruce (*Picea abies*). The authors built different tiny houses in the same form to simulate how the wood would be exposed, and made tests in several European countries with different climates: in Žiri (Slovenia), Ljubljana (Slovenia), Hannover (Germany), Skellefteå (Sweden), and Madrid (Spain) since October 2015. In this study, they compared wood degradation in the five different locations. They concluded that all woods suffer from color changes, mold and stain growth, and moisture content changed after one year of exposure. They noticed that the wood samples presented the lowest moisture content in Madrid, while Žiri and Ljubljana presented the highest moisture content. In all locations, thermally modified wood showed lower moisture contents than those in unmodified timber. Color changes at each site were due to the different weather conditions, with precipitation playing a significant role. In locations where the amount of rainfall was higher, mold growth was also higher.

Tomak et al. [42] evaluated the surface properties of thermally modified wood during 48 months of natural weathering. They worked with thermally modified ash (*Fraxinus excelsior*), iroko (*Milicia excelsa*), Scots pine (*Pinus sylvestris*), and oriental spruce (*Picea orientalis*). The samples were thermally modified at 190 °C and 212 °C for 90 min. The natural weathering took place in Trabzon, Turkey, during the period from October 2011 to

October 2015. At the end of the study, it was concluded that surface roughness increased as the weathering time increased, surface quality and color stability were enhanced with the thermal modification for all wood species, and that better results were achieved with hardwoods rather than softwoods. FTIR data showed that the surface composition of thermally modified wood and unmodified wood were very similar in the first weathering exposure period. They also concluded that thermal modification might not adequately protect the surface appearance and color stability in long-term outdoor conditions.

Humar et al. [54] studied a combined effect of wetting ability and durability. They exposed seven different wood samples: Norway spruce (*Picea abies*), European larch heartwood (*Larix decidua*), European beech (*Fagus sylvatica*), European ash (*Fraxinus excelsior*), Scots pine heartwood and sapwood (*Pinus sylvestris*), sweet chestnut (*Castanea sativa*), and European oak heartwood (*Quercus* sp.). Both untreated and treated wood samples were exposed. The authors used the same little house used by [54], but only the decking specimens were studied. These wood samples had been exposed for five years in Ljubljana (Slovenia). They studied the effect of moisture in wood, constantly comparing it with the various treatments. They related the moisture content with wood decay, and a mathematical model was developed. This model helped to predict the susceptibility of wood to biological attack, depending on moisture content.

In general, modified wood presented a better performance when exposed to weathering. Some wood species were more sensitive to thermal modification, which was reflected in their weathering performance.

Several countries have a long coastline, the effects of which substantially impact the durability of metallic or polymeric materials, consequently impacting their lifespan. One example of this is Portugal [63,64]. The main characteristics of this environment are salinity, wind-driven abrasion, and high solar radiation. Even though different wood constructions exist, no study was found for this environment that compared it with other settings.

In industrial environments, there can be an increased presence of polluting gases and particulate matter. For metallic materials and metallic coatings, the study of environments with $H_2S$, $SO_2$, and $NO_x$ gases is widespread because these contaminants contribute significantly to the degradation of metallic materials [62,65]. Regarding timber, no studies appear to have focused on the influence of industrial and maritime environments on the durability of wood.

### 3.2. Artificial Weathering

Artificial weathering involves exposing test specimens to an artificial UV light source in a cabinet where the temperature, humidity, and water spray are controlled. The problem with this approach is in determining the synergistic effects (interactions) between the different parameters of the weathering process. The critical parameter in all accelerated weathering apparatus is the UV light source, ideally simulating solar radiation.

The artificial weathering or accelerated aging tests allow an estimation of the potential long-term serviceability of materials. They help researchers to understand the chemical reactions involved during degradation [66]. Some weathering conditions can be simulated in the laboratory using climatic chambers to accelerate certain natural weathering conditions.

In terms of wood, the most common accelerated aging test is for UV radiation resistance, humidity, and temperature variations that simulate the wetting and drying periods that occur in nature [40,57–62,67].

The standards ASTM G155 and EN 927–6 are those most commonly used by several authors to evaluate the radiation resistance of materials [53,55,58,59]. ASTM G155 is for polymeric materials and uses xenon lamps, while ISO 927–6 is for wood coatings and uses UV lamps. Other authors did not follow any specific standard (for example, Ayadi et al. [58]). They exposed ash (*Fraxinus* sp.), beech (*Fagus sylvatica*), maritime pine (*Pinus pinaster*), and poplar (*Populus* sp.) to UV light at 340 nm directly for 835 hours. Each cycle included 2h 30 min of irradiation at 60 °C, followed by 30 min of condensation at 50 °C. Srinivas and Pandey [62] also did not follow any specific standard when they exposed

thermally modified rubberwood (*Hevea brasiliensis*) samples to a 1000-Watt xenon arc light source at 50% relative humidity over a period of 300 h. There is no specific standard for thermally modified wood, and the best solution is to use these standards to obtain comparative results.

In all UV resistance studies found in the literature, the researchers evaluated the color stability of thermally modified wood under artificial weathering, using UV radiation at different exposure times (between 75 and 835 h).

In one of the studies found in the literature (Domingos et al. [53]), the authors compared artificial weathering to natural weathering. The authors tried to predict how much time in the UV chamber would be needed to simulate the natural exposure of between six months (75 h) and five years (750 h), using the maritime pine (*Pinus pinaster*). They concluded that 75 h of UV radiation exposure in a climatic chamber did not correspond to the surface color change from 6 months of natural weathering. They noticed that the lightness stayed approximately constant for unmodified wood, and modified wood did not significantly change during artificial weathering. Roughness increases at the beginning of the artificial weathering process until 300 h had passed for unmodified timber, remaining constant. Modified wood demonstrated a very similar behavior but with a higher increase. Glossiness decreased slightly in unmodified wood, with insignificant changes for modified wood.

Some researchers characterized wood samples by SEM [55,59] and FTIR [55,57,59–62] to evaluate the photodegradation caused by UV radiation. SEM showed that some cracks appeared during the degradation. As a result, it was easier for water to enter the cell wall, increasing the wettability. Artificial weathering on the cell wall of unmodified wood was more significant than in modified wood. FTIR data showed that thermally modified wood's $OH/CH_2$ ratio was inversely proportional to the contact angle. The lignin was the most sensitive compound of modified wood samples in the weathering degradation process [55,59,62]. The authors also concluded that thermal modification increased the lignin and crystallized cellulose contents, which can help against weathering. The increase of lignin is as a result of the acid depolymerization of hemicellulose. They also noticed that crystallized cellulose transformed into amorphous cellulose due to artificial weathering, which increases hydroxyl accessibility and, consequently, increases thermally modified wood wettability. The authors noticed a change in color for both modified and unmodified wood in reference studies. FTIR showed that modified wood suffers lignin degradation due to UV light. This study reveals that thermal modification is ineffective for restricting light-induced color changes and the photodegradation of wood polymers.

In general, the chemical and structural modifications found by the different researchers were very similar in wood samples exposed to UV lamps or xenon arc light.

Concerning the purpose of this review, no studies or even standards were found using another artificial weathering factor, such as pollutant gases exposure ($SO_2/NO_x$) and salt spray chambers, to simulate industrial and maritime environments. It is known that seawater can be corrosive to wood [68,69]. Even though the salt spray test is used in coating technologies [70], no study was found for coastal areas with a prevalent wind-driven salt spray.

These studies, conducted by different researchers, helped us to understand which abiotic factors can contribute to wood surface degradation, even considering that the different environments studied are very similar. There is a lack of information for different backgrounds and the effects of long-term weathering exposure for the general use of wood applied at outdoor applications.

## 4. Conclusions

This review allowed us to draw the following conclusions:

1.　The main factors with a significant impact on thermally modified wood degradation were moisture content and UV radiation. The UV radiation promotes color loss and photodegradation of the wood surface, leading to cracks in the wood structure in

both unmodified and modified wood. The moisture content promotes mold, blue stain, and fungal growth, affecting the wood color.

2. The advantages and disadvantages of thermally modified wood were identified. Benefits were the improvement of dimensional stability with thermal modification, promoting some weathering protection. The disadvantages were that thermal change was not beneficial in some wood species, and the modified wood can be sensitive to UV radiation.

3. Species are structurally different, with notable performances in particular environments. It will be essential to consider more studies with a wider range of species (temperate and tropical) and environments, for example, in industrial and maritime environments where the effect of the salinity and pollutant gases are very aggressive for metallic materials; the same effects are unknown for modified wood. In addition, more studies on thermally modified wood are needed for wooden constructions in coastal areas (urban centers with a higher population density). It is necessary to conduct long-term weathering exposure to know how thermally modified woods will be affected by atmospheric contaminants and to predict their behavior under current climatic change scenarios.

4. More field and laboratory tests, including different thermal modification settings and weathering factors, should be conducted to identify which environmental parameters affect wood the most.

The lack of information mentioned in our review can be an opportunity for future work; that is, to understand the degradation mechanisms caused by the weathering factors and to search for other forms of wood protection to promote the improvement of its lifetime service in various weathering contexts.

**Author Contributions:** Manuscript writing, D.G.; manuscript review, S.d.O.A., T.Q., T.D., J.G.; supervision, T.Q., T.D., J.G. All authors have read and agreed to the published version of the manuscript.

**Funding:** This research was funded by FCT (Fundação para a Ciência e Tecnologia, Portugal) by financing the Forest Research Centre (UIDB/00239/2020). Solange de Oliveira Araújo was supported by FCT through research contract (DL57/2016/CPI382/CT0018). Delfina Godinho was supported by FCT through PhD fellowship (PD/BD/142987/2018).

**Acknowledgments:** The authors would like to thank the Fundação para a Ciência e Tecnologia, Portugal, for funding this work and Duarte Neiva for making a final adjustment in the manuscript.

**Conflicts of Interest:** The authors declare no conflict of interest. The funders had no role in the study's design, in the collection, analyses, or interpretation of data, in the writing of the manuscript, or in the decision to publish the results.

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
