# Peer review of "Thermally Modified Wood Exposed to Different Weathering Conditions: A Review"

_forests, doi:10.3390/f12101400_

Round 1

Reviewer 1 Report

Dear Authors,

I have read the article entitled „Thermally modified wood exposed to different weathering conditions: a review:”.

In this work a well-defined review on the thermally modified wood when exposed to natural and artificial weathering is presented. The lack of studies on several specific conditions is pointed out and this could be the start for future research work.

The subject of the study falls within the scope of the Journal. The title and abstract reflect the content. The introduction initiates the reader on the subject and a well-selected list of references was used. All figures and tables are mentioned in the text.   

The paper is ready for publication with some revisions. I have only a few observations.

Page 2 lines 68-70: please split the sentence

Page 2 line 82: …..as it is presented in the review performed by Sandberg et al [18]

Page 7: the authors may add a picture compilation of the treated samples found in the literature, if possible.

Page 8 line 213: This happens because……[54].

Page 8 lines 214-215: the formation of ……occurs.

Page 9 line 251: a mathematical model has been developed.

Page 9 lines 254-255: please rephrase

Page 9 lines 275-278: the sentence is too long

Page 9 line 286: other authors did not follow any standard. This is the case of Ayadi…who exposed….wood species must be mentioned

Page 9 line 289: Srinivas and Pandey [60] did not follow any specific standard. They exposed…wood species must be mentioned.

Page 10 line 296: The studied wood species are:…..

Page 10 lines 300-304: please rephrase

Page 10 lines 324-326: please reconsider the text

Page 11 line 341: ……no studies were found.

Page 11 lines 349-353: split and rephrase

Pages 5, 6, 7: references 36, 37 and 39 are missing.

I hope my revision was of help.

Reviewer 2 Report

The review needs some improvements before publication:

Major:

  • There is plenty of review articles dealing with weathering of thermally modified wood. You should clearly provide the novelty of your review.
  • The first part of the abstract is very general, please modify it. Please focus more on the results of your research.
  • Parts 1.1-1.3 are also very general, please focus more on the results of research in the last 2 decades in this area. Same for other parts.
  • After reading the Abstract, I was expecting more discussion about what you mentioned in lines 19-22 in the Abstract.

Minor:

Lines 26-33: This part is too general, I suggest omitting it.

Lines 37-39: Please add some reference to your statements.

LIne 45: Please explain what do you mean by "other arrangements".

Line 87-89: Please revise the sentence, as it is not completely true.

LIne 90: You are mentioning a range from 180-260, in Table 1, there are also temperatures below 180.

Table 1: Please explain abbs. FWD and OHT as first time used (OHT explained later in line 103, FWD not et al)

Line 113: not all properties you mentioned in lines 114-115 are mechanical properties, please revise.

Table 3: Please use MPa not Mpa for modulus 

Table 3: Please be more precise: Ash (Fraxinus excelsior L) etc. (please check whole manuscript for Latin names). Also please add errors for MOE and MOR: 7760(850.94) MPa, etc., and the significance of the difference (for example for Iroko there was no significant difference in control and HT sample).

Table 3: Please add Latin names for samples from reference 32.

Please consider if Figure 1 is needed.

Lines 151-161: Please be more precise about the definition and main properties of weathering, UV exposure, etc.

Table 4: Please check comments for Table 3

Line 295: not all use 340nm, please revise

LIne 324: "They" you mean authors? :)

Reviewer 3 Report

Dear authors,
can you finally predict how dynamically changing environmental conditions (global warming) will affect the durability of wooden buildings? In which climatic zones (taking into account extreme weather conditions such as strong winds, floods) should special attention be paid to additional wood protection. In part, it has already been mentioned that extreme drought can cause cracking, which in turn causes moisture and fungal growth. However, it would be good to relate this to specific regions, e.g. are timber buildings in southern or northern Europe more prone to deterioration? Perhaps draw conclusions based on studies done in Spain, Slovenia, Germany etc.
It would also be good to contrast the advantages and disadvantages of thermally modified and non-thermally modified wood. After reading the first part of the article and the tables, I got the impression that thermally modified wood is more resistant to moisture and the associated moulds, while in the second part the opposite is true, that the lignin is more easily washed out, which promotes the development of fungi. It has been shown that temperature partially destroys the structure of the wood (core rays), which promotes moisture, and that, on the other hand, the cellulose crystallises and increases resistance to moisture. So is the modified wood more dimensionally stable (as mentioned in the table) or does it get wet more easily (as also mentioned in the field tests)? Is it therefore recommended for use over swimming pools, on terraces and building elevations, or does it need additional protection? The same goes for UV, as modified wood has been shown to discolour faster than unmodified, but there are exceptions like larch. You can put this information together in a table, for example, and after reading it give the reader a clear picture of when and where modified wood was used successfully (which tree species) and when it failed completely (was no different from the control). Thermal treatment requires energy (which is not indifferent to the environment) and is expensive (and energy prices are constantly rising).

Round 2

Reviewer 2 Report

The manuscript was improved, I suggest accepting it.

Author Response

Thank you once again for your revisions.